# Proteo-Transcriptomic Analysis of the Venom Gland of the Cone Snail *Cylinder canonicus* Reveals the Origin of the Predatory-Evoked Venom

**DOI:** 10.3390/toxins17030119

**Published:** 2025-03-02

**Authors:** Zahrmina Ratibou, Anicet E. T. Ebou, Claudia Bich, Fabrice Saintmont, Gilles Valette, Guillaume Cazals, Dominique K. Koua, Nicolas Inguimbert, Sébastien Dutertre

**Affiliations:** 1CRIOBE, USR 3278—EPHE-CNRS-UPVD, Université de Perpignan via Domitia, 58 Avenue Paul Alduy, 66860 Perpignan, France; zahrmina.ratibou@univ-perp.fr; 2ERBB, LaMBB, Institut National Polytechnique Félix Houphouët-Boigny, Yamoussoukro, BP 1093, Côte d’Ivoire; ediman.ebou@inphb.ci (A.E.T.E.); dominique.koua@inphb.ci (D.K.K.); 3IBMM, Université de Montpellier CNRS, ENSCM, 34095 Montpellier, France; claudia.muracciole-bich@umontpellier.fr (C.B.); gilles.valette@umontpellier.fr (G.V.); guillaume.cazals@umontpellier.fr (G.C.)

**Keywords:** cone sails, *Cylinder canonicus*, molluscivorous, venom glands, predatory-milked venom, conotoxins, proteomics, transcriptomics

## Abstract

Cone snails are carnivorous marine predators that prey on mollusks, worms, or fish. They purposefully inject a highly diversified and peptide-rich venom, which can vary according to the predatory or defensive intended use. Previous studies have shown some correlations between the predation- and defense-evoked venoms and specific sections of the venom gland. In this study, we focus on the characterization of the venom of *Cylinder canonicus*, a molluscivorous species collected from Mayotte Island. Integrated proteomics and transcriptomics studies allowed for the identification of 108 conotoxin sequences from 24 gene superfamilies, with the most represented sequences belonging to the O1, O2, M, and conkunitzin superfamilies. A comparison of the predatory injected venom and the distal, central, and proximal sections of the venom duct suggests mostly distal origin. Identified conotoxins will contribute to a better understanding of venom–ecology relationships in cone snails and provide a novel resource for potential drug discovery.

## 1. Introduction

Cone snails are marine mollusks which belong to the large family Conidae. This family groups about a thousand species of gastropod mollusks which possess a venom gland and modified radular teeth [1,2]. These carnivorous predators feed on fish, mollusks, or worms [3]. They produce and use a complex venom mixture intended to paralyze small prey or deter potential predators for defensive purposes [4]. *Conus* snail venoms constitute a rich source of new drug leads due to the modulatory actions of their components towards key physiological receptors such as ion channels, G-protein coupled receptors (GPCRs), and other proteins [5]. The small and highly structured peptides called conopeptides (with one or without disulfide bond) or conotoxins (disulfide-rich) are mainly responsible for the observed biological activity of cone snail venoms, although some small molecules and proteins can also contribute to the envenomation process [6,7].

Most *Conus* species produce highly variable venoms, the origin of which remains unclear due to the interference of multiple factors, such as age, environment, feeding habit, envenomation strategy, etc. [8]. In fact, intraspecific venom variation is known to occur not only within species but also within a single specimen, and even between consecutive venom injections [8,9,10,11]. For instance, some cones have demonstrated the ability to purposefully select different assortments of conotoxins following predatory or defense stimuli. The deadly fish-hunter, *Gastridium geographus*, has displayed chemical and biological differences in predatory and defense venom [4,11]. This observation was directly correlated with the uneven distribution of toxins across the venom gland. For example, previous studies on *G. geographus* and *Pionoconus striatus* have revealed a distinct separation between the nature of conotoxins found in the proximal section of the venom gland, which is closer to the venom bulb, and the distal section of the venom gland, which is nearer to the proboscis, where the venom stings occur [4,12].

Although common, the diversification of cone venoms is an unpredictable phenomenon that occurs at different levels. Therefore, integrating analytical techniques is essential to accelerate the identification of distinct predatory and defensive conotoxin repertoires [9]. For this purpose, multiple omics approaches, or “venomics”, are combined to provide sequence and structural information. First, venom profiling is performed to record a fingerprint of the injected venom and compare it to the dissected venoms. Next, venom gland transcriptomics allows the recovery of all toxin sequences and their classification into gene superfamilies [13]. Proteomics then permits us to identify the conopeptides in each venom sample to establish correlations between the venom gland and the injected venoms. To date, very few studies have delved into venom gland heterogeneity. Among the cones that have been characterized through venomics are piscivorous species such as *Gastridium geographus, Pionoconus striatus*, and *Pionoconus consors*, and vermivorous species such as *Vituliconus planorbis* [14] and *Rhizoconus vexillum* [15]. Molluscivorous species remain understudied.

In this work, we present the whole venom characterization of the mollusk-hunting cone *Cylinder canonicus* (Figure 1) [2]. The shells of cones in the *Cylinder* clade present a “tented” pattern made of white triangles over a brown-to-yellow background. *C. canonicus* resembles the closely related species *C. textile*, and to a lesser extent *C. episcopatus* [16]. Such cone species naturally live in coral reef environments throughout the Indo-Pacific oceans. Within the same clade are also found well-known cones such as *Cylinder victoriae* and *Cylinder ammiralis*, for which evidence of distinct predatory and defensive venoms was reported but possible correlation with specific sections of the duct was not investigated [4,17]. To decipher the origins of the injected venom, we implemented a venomics strategy to the study of the venom of *C. canonicus* by combining transcriptomics and proteomics analyses. Information on the spatial distribution of conotoxins along the venom gland will provide further clues to help understand the complex venom–ecology relationships in cones.

## 2. Results and Interpretations

### 2.1. Venom Profiling: LC-MS Traces of Predatory-Evoked and Dissected Venoms

Overall, predatory-milked venoms collected from the three individuals of *C. canonicus* displayed similar profiles of intermediate complexity (Figure 2). However, a detailed investigation of the detected components revealed notable differences. Indeed, mass profiling showed important variations, with a total of 159 unique masses detected (Figure 2), but only 12 (8%) being shared between the three individuals, which include 839.3, 2114.8, 2158.8, 1024.4, 1453.5, 1454.5, 2995.2, 2913.1, 2940.0 Da, 2740.1, 3041.2, and 3265.3. The major late eluting peaks (between 23.7–25.0 min) include 2913.1, 2927.1, and 2740.1 Da peptides, and their combined area comprises more than a third of the total summed peak areas (30%). Interestingly, one specimen (MV1) injected only two (2913.1, 2927.1) out of these three major peptides.

To determine the origin of the predation-evoked venom, extracts of the venom gland were also analyzed using LC-MS. Clear compartmentalization was observed, with distally- (D and DC) and proximally-dissected venoms (P and PC) presenting drastically different profiles (Figure 3A,B). For instance, peaks detected between 23.0–25.0 min in the distal sections were the most intense ions as they represent 40% relative to the total peak area (1024.4 Da, 2913.1 Da, and 2740.1 Da), while being minor in the proximal sections (3%). In contrast, proximal sections displayed major peaks early in the chromatograms, especially at 8.3 min (3139.9 Da), 8.6 min (2940.0 Da), 9.2 min (3122.9 Da), 15.0 min (2046.5 Da), and 15.5 min (1209.4 Da). These ions correspond to 30% of the total area under the curve (Figure 3B) compared to being present at only 5% in distal sections. When compared to the predation-evoked venom LC-MS trace, it appears that the distal traces are the most similar. In particular, the late eluting components (second half of the chromatogram) show a good correlation in terms of the intensity of the major ions. In the first half of the chromatogram, however, contributions from the proximal parts are also apparent, although not prominent. In total, 239 unique masses were detected throughout the venom gland sections, of which 46 (~20%) were common to proximal, distal, and predation-evoked venoms (Figure 3C).

Overall, dissected venoms were more complex than milked venoms, especially the proximal venoms. In an attempt to gain further information on the contribution of each section of the duct to the composition of the predatory venom, mass profiling was performed on each venom sample by MALDI-MS analyses.

### 2.2. Venom Profiling: MALDI-TOF-MS of Predatory-Evoked and Dissected Venoms

Matrix-assisted Laser Desorption/Ionization mass spectrometry (MALDI-MS) was performed in parallel to LC-MS to acquire a mass profile or a fingerprint of the venom samples [19]. The predatory milked venoms of the three *C. canonicus* specimens, as well as the distal and proximal dissected venoms, revealed great diversity through MALDI-MS (Figure 4). A total of 410 masses in total were detected, of which only 17 (4%) were shared between all samples. Major peaks include peaks at 1453.5, 1739.1, 2093.1, 2112.9, 2990.0, and 2910.0. From the fingerprint profiles (Figure 4), the contribution of each venom duct section to the composition of the predatory venom does not appear as straightforward as the LC-MS results showed. Indeed, although some ions show some clear concordance between distal and predatory venom samples (i.e., 2112.67 Da), other ions are common specifically to proximal and predatory venom samples (i.e., 1739.05 Da).

Mass spectrometry is an efficient strategy to evaluate venom complexity; however, it does not provide information on the identity of the toxins. Primary sequence information is necessary to more precisely decipher the nature of the conotoxins used in predation by *C. canonicus*. This identification will be accomplished through the coupling of proteomic and transcriptomic data.

### 2.3. Transcriptomics: mRNA Transcripts Expression in the Whole Venom Gland of Cylinder canonicus

In total, 98 full conopeptide precursors were extracted from the 252,104 transcripts constituting the entire transcriptome of *C. canonicus*. Precursors were distributed into 22 gene superfamilies, of which the most diversified were M, O1, O2, and conkunitzins in the venom gland tissues (Figure 5). These major gene superfamilies accounted for more than 50% of all conotoxin precursors. Overall, 14 paralogs were classified into the superfamily M (14%), and 12 paralogs for each of the O1, O2, and conkunitzins gene superfamilies (12%). Other gene superfamilies were annotated with lesser paralogs such as the T gene superfamily (8 paralogs), Con-ikot-ikots (7 paralogs), and P gene superfamily (5 paralogs). Finally, minor gene superfamilies often comprise less than four paralogs, including A, B, D, F, H, I2, I3, S, and U, as well as coninsulins, elevenins, conopressins-conophysins, conorfamides, and prohormones. Sequence alignments of all conopeptide precursors according to gene superfamilies may be found in the Appendix A (Appendix A, Appendix A).

### 2.4. Proteomics: Characterization of the Predatory-Evoked Venoms and Dissected Venom Duct Sections

LC-MS and MALDI-MS revealed some levels of diversity in predatory venoms and across the venom gland of *C. canonicus* specimens. Therefore, prior to performing a bottom-up proteomics approach, all three predatory venom samples were pooled (to minimize the intraspecific variations), as well as the D/DC and P/PC sections together (as LC-MS revealed similar composition). Thus, all three venom samples (predatory, D, and P sections) were analyzed by LC-MS/MS, and the resulting fragmentation spectra were processed with bioinformatic tools to match sequences in the database (corresponding to the whole venom gland transcriptome of *C. canonicus*). Consequently, a total of 466 unique transcriptomic sequences were matched and identified from all venom samples. Out of the 466 sequences, more than 85% were found in the distal section of the venom gland (399 sequences), compared to 45% in the proximal section (209 sequences) and 12% in the pooled predatory venoms (56 sequences) (Figure 6A). As expected from the LC-MS profiles, distal and proximal venom gland sections both display a higher complexity than predatory-pooled milking venoms.

Each transcriptomic sequence that was identified was then run through a similarity-based protein BLAST search to identify protein functions or the gene superfamily for the conotoxins. In total, 354 proteins were non-related to the venom function, also qualified as housekeeping proteins (76%), 84 proteins were venom-related proteins (18%), and finally, 25 sequences (5%) were identified as conotoxins precursors (Figure 6B, Appendix A, Appendix A). Housekeeping proteins generally include proteins involved in cell maintenance, constantly required by the cell, such as GAPDH, ribosomal proteins, Ras protein, histones, and elongation factor. They were logically detected in dissected venom gland sections (Appendix A, Appendix A). However, venom-related proteins were found in both milked and dissected venoms. They include proteins that participate in venom production and delivery such as protein disulfide isomerase, carboxypeptidase E, von Willebrand factor A-like, or transmembrane protease serine 2-like (Appendix A, Appendix A). An extensive list of the entire proteome with annotations may be found in the Supplementary Information (Appendix A, Appendix A).

About 5% of the entire bottom-up proteomic data were matched to 25 sequences of conotoxins identified in the milked and dissected venoms. They were distributed across nine gene superfamilies, including O1, T, O2, M, I1, E, S, P, and A, in order of abundance (Figure 6C). Gene superfamilies with the highest amounts of paralog identified were O1 (six peptides), T (five peptides), M, and O2 (four peptides). Fifteen of these sequences had already been identified by the previous transcriptomics analysis. They correspond to one A-conotoxin (Can01), four M-conotoxins (Can21, Can22, and two paralogs of Can28), three O1-conotoxins (Can33-Can35), three O2-conotoxins (Can45, Can50, and Can53), one P-conotoxin (Can55), one S-conotoxin (Can61), and three T-conotoxins (three paralogs of Can62) (Table 1).

Remarkably, proteomics analyses have allowed us to uncover 10 additional conotoxins that were not present in our annotated transcriptome: four O1- (Can102, Can103, Can104, and Can105), two T- (Can107, Can108), two I1- (Can100 and Can101), one O2- (Can106), and one E-conotoxin (Can99) (Table 1). The main reason for this discrepancy lies in our stringent criteria for the selection of our transcriptomic sequences: full precursors containing the signal peptide, starting with a methionine and ending with a stop codon. Indeed, among the 10 “missed” sequences, 9 were incomplete precursors (either missing the N- or C-terminal part and therefore rejected in our bioinformatic analysis). Yet, this result is highly significant given that 40% of the proteomically validated conotoxin sequences were not initially listed in our retrieved precursors, showing the importance of combining both transcriptomics and proteomics for the most complete venom coverage. For instance, the conotoxin sequence of O1-Can102 (full contig sequence is 176 amino acids long) demonstrated the largest amount of peptide-spectrum matches (PSM) within predatory and distal venoms (Figure 7A,B), and remained also major in the proximal venoms.

In addition to the validation of transcriptomic sequences, proteomics uncovers a complementary layer of diversity in the form of post-translational modifications (PTMs) and other modifications involved during the gene-RNA-protein translation processes. PTMs and other modifications that are involved in venom production have been assumed to play an important role in improving peptide activity and/or stability [20]. The most common PTMs identified in the proteomics of all samples, outside of the carbamidomethylation (due to sample preparation), include hydroxylation and carboxylation (Figure 7C). Furthermore, deamidation of asparagine (Asn) and glutamine residues (Gln) was also frequently detected (Figure 7C).

Additionally, a diversity of cysteine frameworks is observed in the conotoxins identified through proteo-transcriptomics (Table 2, Appendix A, Appendix A). First, the VI/VII cysteine framework (C-C-CC-C-C) was the most represented in all conotoxins, especially in the gene superfamily O (O1, O2, and O3). This arrangement was also identified in a few M- and U-conotoxins. The framework IX was found in a few O1-, conkunitzins, and P gene superfamilies. Additionally, a few frameworks were only encountered in specific gene superfamilies such as framework III (CC-C-C-CC) in M-conotoxins, framework V (CC-CC) only in T-conotoxins, framework VIII (C-C-C-C-C-C-C-C-C-C) in S-conotoxins, framework IX (C-C-C-C-C-C) in P-conotoxins, or even framework XI (C-C-CC-CC-C-C) only in I-conotoxins (I2, I3). Many other conotoxins displayed no cysteine or the classification is unknown. A few other sequences, especially those that were identified through proteomics, were missing a fragment of the mature peptide, making identification difficult, such as the conotoxins I1-Can101, O1-Can103 and O1-Can104 (Table 1) (Appendix A, Appendix A).

## 3. Discussion

### 3.1. Characterization of Cylinder Canonicus’ Venom Through a Venomics Approach

In this study, the venom composition of the molluscivorous *C. canonicus* was extensively uncovered through a venomics approach. The composition of the predatory venom and venom gland sections was analyzed by mass spectrometry, transcriptomics, and proteomics. First, mass spectrometry-based analyses demonstrated the complex and diverse nature of *C. canonicus* venom. Both MS techniques have allowed us to detect around 400 different masses, with a total of 399 masses through LC-ESI-MS and 410 through MALDI-MS. LC-ESI-MS chromatograms revealed comparable profiles for predatory venoms collected from different specimens of *C. canonicus* (Figure 2). Although the general profile is preserved, we noted different levels of complexity between the specimens. These results are consistent with previous works, which have demonstrated intraspecific variation between specimens of the same *Conus* species, such as *C. consors* [11], *C. striatus, C. catus* [8], and *C. geographus* [21]. In particular, the predatory venom of specimen 4 was more complex than that of specimens 2 and 1. Yet, the major components (i.e., 2158.8, 2913.1, 2927.2, and 2740.1 Da) were conserved among all three specimens. This suggests that this species selects a very precise cocktail of toxins for prey capture.

### 3.2. Effect of the Instrumentation Chosen for Venom Profiling

Differences in instrumentation and chemistry between LC-ESI-MS and MALDI-TOF MS allow the profiling of different venom components. Indeed, previous venomics studies have revealed the complementarity of integrating different approaches for venom profiling [22,23]. Overall, the high diversity between predatory milked and dissected venoms was confirmed again by MALDI-MS. Ion detection performed in reflectron mode allows increased resolution of peaks; however, the sensitivity is inversely affected [19]. Contrary to LC-MS analyses, sample mixtures are not separated prior to MALDI-MS. This is associated with the ion suppression effect, caused by different parameters such as the nature of the matrix. As a result, major components might be detected to the disadvantage of minor components, which become undistinguishable from noise [24]. For this reason, LC-MS results were more revealing in the context of the predatory venom’s origin in the venom gland.

### 3.3. Compartmentalization of the Venom Gland vs. Predatory Venom

The profiling of the venom gland of *C. canonicus* demonstrates a compartmentalization of venom content across the duct. Indeed, two separate profiles are observed between the distal and proximal sections of the venom duct. This compartmentalization has been suggested to be the underlying mechanism behind the evolved ability of some species to inject different predatory and defensive venoms, as proposed in previous studies [4,12,21]. Overall, the separation between each section of the gland seems consistent with the possibility of two distinct venom compositions arising from distal and proximal parts. Yet, some components show a continuous distribution across the duct but with varying intensities. When LC-MS traces are compared, the origin of the predatory venom appears to be largely contributed by the distal part. However, closer investigation shows that the predatory venom contains a blend of mass peaks from the entire venom gland. If the intensity of the ions could be considered, then the resemblance between distal and predatory venoms would become more apparent. Therefore, the comparison between the venom gland and the milked venoms should be assessed relative to the amount of each component. From our results, the venom injected for prey capture appears selected in the venom gland portion that is closer to the proboscis (distal section). To better understand the prey capture strategy of *C. canonicus*, the biological and structural characterization of these predatory conotoxins on prey tissues will be the focus of further studies.

### 3.4. Venom Characterization Through Transcriptomics and Proteomics

Combined proteo-transcriptomics successfully elucidated 550 protein sequences, including 354 housekeeping proteins, 84 venom-related proteins, and 108 conotoxin precursors. “Housekeeping” proteins are considered to be ubiquitous, i.e., present in every cell tissue. They include proteins required for cell maintenance such as RNA translation (ribosomal proteins), degradation (ubiquitin chains), folding (chaperones), cellular structure (actin-like), or transport (vesicle-trafficking). They are expressed consistently, in a measured amount to allow proper cell functioning [25]. Therefore, their high expression in the venom gland over the milked venom is coherent. Unlike “housekeeping” proteins, venom-related proteins regroup various types of proteins that are involved in venomous functions. These proteins were detected both in the venom gland but also in the predatory venom of *C. canonicus*. They include various enzymes and proteins that allow the venom gland to produce and mature venom precursors (conotoxin/conopeptide precursors) but also respond to signals that initiate venom release and delivery into prey [26]. We identified for example an angiotensin-converting enzyme-like protein (ACE), a von Willebrand factor A-like protein (VWF), other metalloproteinases, carboxypeptidases, protein disulfide isomerases (PDI), puromycin-sensitive aminopeptidase-like (metal-lopeptidase involved in protein degradation) [27], polyubiquitin-like proteins (in-volved in degradation and recycling of proteins). For instance, the ACE is known to catalyze the production of angiotensin II, involved in blood pressure regulation, but its implication in predation is unknown [28]. The VWF-like protein was also previously identified in another cone of the same subclade, C. ammiralis, but its function has not established yet. It is known to interfere with platelet aggregation [17] (Appendix A
Appendix A).

### 3.5. Identification of the Major Constituents of the Predatory-Evoked Venom

Through our combined proteo-transcriptomic study, 16 conotoxin sequences were identified in the predatory venom of *C. canonicus* (Appendix A, Appendix A). Most represented by far is the superfamily O1, with five conotoxin sequences detected. This is highly relevant to the prey capture strategy, given that related peptides found in the venom of *C. textile* were found to produce potent effects in mollusks. In particular, King-Kong, TxIA, or TxVIA (the same peptide identified by several groups, hence different names) was found to be the most potent mollusk-specific toxin in *C. textile* venom, and it resembles the sequence Can033 [29]. Indeed, the most intense ion detected in the LC-MS of the predatory venom of *C. canonicus* corresponds to the mature conotoxin of sequence Can33. The next most intense ion is Can102, which also belongs to the same family. Based on the strong contractive effect on limpet foot muscles found for TxVIA, we can expect that similar activity of Can33 and Can102 would rapidly lead to the incapacitation of the prey [29]. Three O2 sequences (Can45-Can50-Can53) were also identified, with one (Can50) resembling Gla(3)-TxVI, another peptide identified in *C. textile* venom. Unfortunately, the biological activity of the O2 superfamily of conotoxins on prey-specific tissues is unknown; hence, we cannot hypothesize their possible role in prey capture. Similarly, no specific role can be attributed to the 3 T superfamily conotoxins found in the predatory venom of *C. canonicus*, as biological characterization is also lacking for this family, as well as for the single M, A, S, P, and E conotoxins found. However, the presence of these conotoxins strongly suggests an essential ecological role in predation, and further physiological and pharmacological characterization of mollusk tissues/receptors is warranted.

## 4. Conclusions

To conclude, this work presents a comprehensive characterization of the venom produced by *Cylinder canonicus* through proteo-transcriptomics analyses. We identify 108 conotoxins belonging to 24 gene superfamilies, especially M, O1, O2, conkunitzin, and T. In correlation with this, we have characterized some of the major components of the predatory venoms as being O1-conotoxins. Our results also suggest that *C. canonicus* selects the conotoxins to be injected into their predatory venom, mostly in the distal part of the venom gland. We also demonstrate compartmentalization of the venom gland, which correlates to previous works, and the ability to produce two types of venom (predatory and defensive). This is a valuable contribution as it provides new transcriptome and proteome data that can be useful for a better understanding of cone snail venom ecology. In further work, we will attempt to obtain defensive venom to complete the study and better understand the purpose of the compartmentalization of the venom gland.

## 5. Materials and Methods

### 5.1. Cone Snail Collection and Venom Extraction

Specimens of *Cylinder canonicus* were collected in February 2023 from the barrier reef surrounding the Island of Mayotte in the Mozambique channel under national and local permits/authorizations. Cone snails were then acclimated and maintained in marine aquariums in the laboratory. Specimens of *C. canonicus* were fed weekly or fortnightly locally collected gastropod mollusks of the family Nassariidae (*Tritia sp.*). Milked venom was obtained from three different *C. canonicus* specimens using previously described methods [30] and venom gland extracts (sections) and tissues (RNA) were retrieved from two other specimens. Predatory-evoked milking venoms were obtained by luring cones with the live mollusk prey. Once the cone extends its yellow proboscis out, we intercept the sting with a collecting Eppendorf tube, lined with a fine slice of another prey’s foot over a parafilm. Milking venoms were translucid with a small white precipitate at the bottom of the tube. A whole venom gland was used for mRNA extraction, and a second venom gland was dissected into four sections (distal, distal-central, proximal-central, and proximal) before extracting the venom content for mass spectrometry and proteomic analyses. Finally, predatory-evoked venoms were milked from three specimens of *C. canonicus* (MV1, MV2 and MV4). Predatory behavior was initiated by facing a Nassariidae mollusk to each *C. canonicus* specimen and intercepting the injection with an Eppendorf tube covered with parafilm and a thin layer of prey mollusk foot. Milked samples were lyophilized and stored at −20 °C before use.

### 5.2. RNA Extraction and Sequencing

One *Cylinder canonicus* specimen was dissected on ice to extract the venom gland in the form of a long and thin duct. The venom duct was placed in 1 mL of TRIzol reagent (Invitrogen, Carlsbad, CA, USA) in an Eppendorf tube and the purified total ribonucleic acid (RNA) was extracted by following the manufacturer’s instructions [12]. Next, mRNA was purified from the retrieved total RNA using the “Oligotex mRNA Mini Kit” (Qiagen, Valencia, CA, USA). Finally, mRNA extracts were submitted to Montpellier GenomiX (MGX, BioCampus Montpellier, CNRS) for sequencing. Complementary DNA (cDNA) libraries were constructed and sequenced using a high-throughput Illumina sequencer (Illumina Inc., San Diego, CA, USA) with a TruSeq Stranded mRNA Sample Prep Kit and a TruSeq Library indices following the manufacturer’s instructions. Paired-end sequencing (2 × 150 bp) yielded several millions of short-read sequences after filtering poor-quality reads. cDNA short raw reads were controlled for quality using FastQC v.0.11.9 (http://www.bioinformatics.babraham.ac.uk/projects/fastqc), trimmed, then assembled into longer contigs using Trinity v2.13.2. The assembled contigs were then translated into amino acids (six reading frames) in silico [17].

### 5.3. Transcriptome Annotation

Conotoxin sequences were then retrieved using, in parallel, a manual and automatic approach. First, a locally-built transcriptomic platform (VenUM) was used to manually extract conopeptide precursors from the venom gland transcriptome of *Cylinder canonicus*. Registered conopeptide precursors found on ConoServer (www.conoserver.org) from species of the same clade, such as *C. victoriae* and *C. textile*, were used as queries to search against the transcriptome of *C. canonicus* to extract and annotate conopeptide precursors. Additionally, the ConoPrec tool from ConoServer was employed to identify cysteine frameworks, cleavage sites, and mature sequences [31]. For further validation of conopeptide superfamilies and other protein annotations, protein BLAST searches were conducted using BLAST+ v2.16.0+ [32] and the non-redundant protein sequences and Swiss-Prot r2024_03 databases [33,34]. To classify gene superfamilies automatically, the bioinformatic tool ConoDictor v2.4.1 was utilized (https://github.com/koualab/conodictor) [35]. This tool enables sequence classification and predicts superfamily assignments [36]. Finally, multiple conopeptide sequence alignments were generated using the open-source software Jalview v.2.11.4.0 (www.jalview.org) and ClustalW v2.1 with default parameters (www.clustal.org/clustal2/) [37,38].

### 5.4. Mass Spectrometry (MS)

**LC-ESI-MS.** Crude lyophilized venoms were resuspended in water and concentrations were evaluated by a NanoPhotometer^®^ (Implen GmbH, Munich, Germany). Electrospray ionization mass spectrometry coupled with liquid chromatography (LC-ESI-MS) experiments were performed following previous protocols [17]. Separation by liquid chromatography was performed on an Acquity H-Class ultrahigh-performance liquid chromatography (UPLC) system. Around 20 µg of each venom sample were injected onto a Kinetex C_18_ 100 Å column (2.1 mm × 150 mm, 3 μm) (Phenomenex, Torrance, CA, USA) fitted with a precolumn for separation at 0.4 mL/min in a 0–80% gradient of a solvent B during 60 min (A: water (H_2_O) + 0.1% formic acid (FA); B: acetonitrile (ACN) + 0.1% formic acid). ESI-MS analyses were performed on a Waters Synapt G2-S (Waters Corporation, Milford, MA, USA) equipped with a time of flight (TOF) detection cellule and was used in positive mode across a masse range of 1000–3000 Dalton (Da) and scan time of 1 s. Each sample was infused in the ionization source at a flow rate of 5 µL/min. Then, ionization was performed with a capillary tension of 3 kV. Temperatures were set at 120 °C for the source and 350 °C for the desolvation gas. Gas flows were regulated at 850 L/h for the desolvation gas and at 6 L/h for the nebulizing gas. Total ion current (TIC) chromatograms and molecular masses were processed with Mass Lynx software (version 4.1, Waters, Corp., Milford, MA, USA).

**MALDI-TOF-MS.** MALDI-TOF mass spectra were acquired using a RapifleX^®^ MALDI mass spectrometer (Bruker Daltonics, Billerica, MA, USA). All samples were diluted at 0.10 mg/mL. In parallel, a solution of α-cyano-4-hydroxycinnamic acid (HCCA, ACROS Organics, USA) matrix was saturated at 10 mg/mL by dissolving 10.0 mg of HCCA in 1 mL of 30% ACN in aqueous 0.1% TFA (70:30:0.1, ACN/H_2_O/TFA). Samples were spotted on an MTP 384 polished steel plate (Bruker Daltonics, Billerica, MA, USA) according to the dried-droplet spotting technique which consists of mixing 1 µL of the sample with 1 µL of HCCA matrix (α-Cyano-4-hydroxycinnamic acid) and spotting 1 µL of each sample in two different spots. External calibration was performed using a mixture of standard peptides (Bradykinin 757.399 Da, Angiotensin II 1046.542 Da, Angiotensin I 1296.685 Da, Substance P 1347.735 Da, Bombesin 1619.822 Da, ACTH clip 2093.086 Da, ACTH clip 2465.198 Da, Somatostatin 3147.471 Da). Mass spectrometry spectra were acquired in positive ion mode at a frequency of 5000 Hz, by applying 6000 shots/per sample spot (4 × 1500 shots) with a laser power of 30%. Ion detection was performed on a time of flight (TOF) detector in reflectron mode with a mass range of 500–5000 Da. FlexControl 3.0 software was used for data acquisition and FlexAnalysis 4.0 (Bruker Corporation, Billerica, MA, USA) for data treatment.

### 5.5. Proteomics

The milked predatory venom of three specimens of *Cylinder canonicus*, as well as dissected venoms of distal (D and DC) and proximal (PC and P) sections of the venom gland were prepared prior to proteomic analysis as described [17]. A total of 50 µg of pooled milked (all 3 predatory venoms), distal (D + DC), and proximal (PC + P) venoms were first diluted in 89 µL of triethylammonium bicarbonate (TEAB) 100 mM under stirring during 30 min at room temperature. Disulfide bonds were reduced with dithiothreitol (DTT) at 10 mM for 30 min at 60 °C and alkylated with iodoacetamide (IAA) at 50 mM before 30 min incubation in the dark at room temperature. The venom samples were then enzymatically digested with 1.5 µg Trypsin (Gold, Promega, Madison, WI, USA) and incubated overnight at 30 °C. Samples were desalted on OMICS C_18_ tips (OMIX Tips C_18_ reverse-phase resin, Agilent Technologies Inc., Santa Clara, CA, USA) for purification and concentration of peptide/protein content, then dehydrated in a vacuum centrifuge.

Venom samples were analyzed in nano-flow liquid chromatography coupled to tandem mass spectrometry (Nano-LC-MS/MS). Samples were resuspended in 20 μL of buffer A (0.1% formic acid), and 1 μL was loaded onto an analytical reversed-phase column (250 × 75 mm, Acclaim Pepmap 100 C_18_, Thermo Fisher Scientific). Samples then were separated with an Ultimate 3000 RSLC system (Thermo Fisher Scientific, Waltham, MA, USA) coupled to a Q Exactive HF-X instrument (Thermo Fisher Scientific, Waltham, MA, USA) via a nano-electrospray source (nanoESI), using a 123 min gradient of 6–40% of buffer B (80% ACN, 0.1% formic acid) and a flow rate of 300 nL/min [17].

Raw data were loaded into PEAKS^®^ Studio 8 software (Bioinformatics Solutions Inc., Waterloo, Canada). Precursors and fragment ions were identified with a monoisotopic mass tolerance of 0.1 Da. Carbamidomethylation was set as the potential post-translation modification to be present. The full venom gland transcriptome of *Cylinder canonicus* was used as a database for peptide/protein sequencing. After spectra processing and protein identification, sequences were filtered by applying an FDR (False Discovery Rate) of 1%, at least 2 unique peptides were set to confirm a protein, and a de novo only ALC (average local confidence) score of 80%.

### 5.6. Identification of Conotoxin in Venom Samples

Proteomics and transcriptomics conotoxin results were further exploited to identify conotoxin detected in LC-MS of predatory venoms. ConoServer tools ConoPrec and ConoMass were used for mass prediction (http://www.conoserver.org) [39]. First, the FASTA file of all conotoxin precursor sequences obtained from proteo-transcriptomics was uploaded to the ConoPrec tool (http://conoserver.org/?page=conoprec) to predict conotoxin mature peptides. Mature peptides were then loaded in the tool ConoMass 1 for mass computation (http://conoserver.org/?page=ptmdiffmass). This tool is useful for complex samples such as cone venom as it provides a prediction of the monoisotopic and average masses of each mature sequence, including the corresponding sequence and the number and type of post-translational modifications (PTMs) [39]. Mass predictions were performed on non-reduced conotoxin sequences, including PTMs such as N-terminal amidation, pyroglutamylation, proline hydroxylation, and tryptophane bromination. Computed masses were then matched with an LC-MS experimental mass list for each sample.

## Figures and Tables

**Figure 1 toxins-17-00119-f001:**
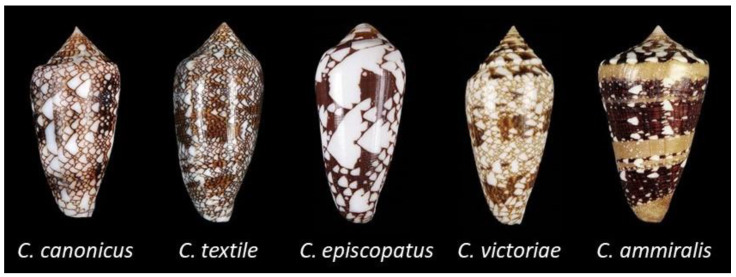
Shells of molluscivorous cones that belong to the same clade (subgenus) *Cylinder*.

**Figure 2 toxins-17-00119-f002:**
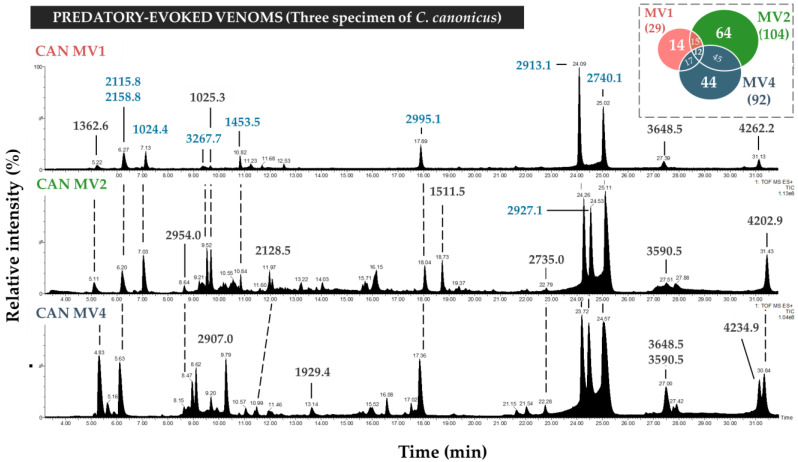
LC-MS total ion current (TIC) chromatograms of predatory-evoked venoms from three different specimens of *Cylinder canonicus*. TIC chromatograms represent a summed intensity of all ions detected on the mass spectrometer detector depending on the time (minutes) and are normalized according to the most intense peak (relative intensity %). Chromatogram peaks were annotated with the molecular monoisotopic base peak mass (most intense peak on the mass spectrum in Dalton, Da). In this figure, molecular masses of the largest peaks in predatory venoms were colored blue. The Venn diagram on the top right represents the complexity and diversity of predatory-evoked venoms of three specimens of *C. canonicus* (MV1, MV2, MV4).

**Figure 3 toxins-17-00119-f003:**
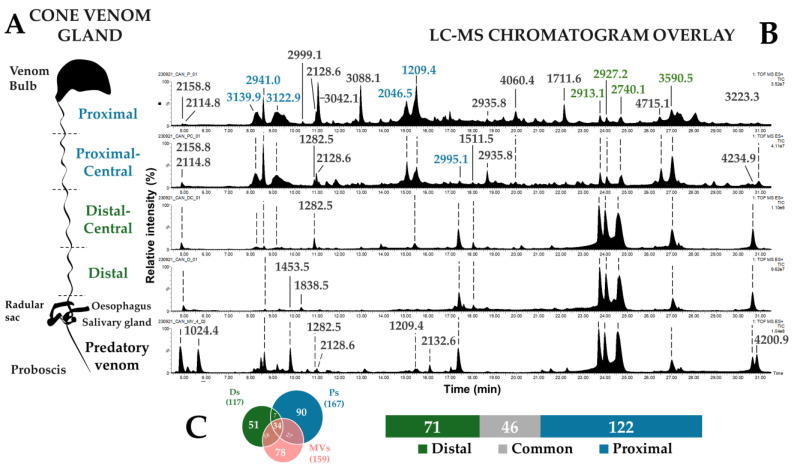
Venom diversity across the venom gland. (**A**) Schematic representation of the venom gland of a cone snail adapted from Prashanth et al. (2016) [15]. The venom bulb functions as a muscle that allows the venom to be expelled by muscle contraction through the venom gland. The venom duct was dissected in four separate sections: the proximal and proximal-central sections, which are closer to the venom bulb, and the distal-central and distal sections, which are further from the bulb and closer to the proboscis end, where venom injection occurs [4,18]. (**B**) LC-MS chromatogram overlay of dissected venoms extracted from four different sections of the venom gland and a predatory venom (CAN MV4) of a cone snail, *Cylinder canonicus*. Chromatogram peaks were annotated with the molecular monoisotopic base peak mass (most intense peak on the mass spectrum in Dalton, Da). In this figure, molecular masses of largest peaks in the distal and proximal venoms were respectively colored green and blue. (**C**) Diagrams illustrate mass distribution across the venom gland. The Venn diagram on the bottom-left represents shared masses between pooled predatory CAN MV1 + MV2 + MV4 milking venoms (MVs) and pooled dissected distal D + DC (Ds) and proximal P + PC (Ps) venoms. Proximal venoms display higher number of detected masses, followed by milkings and Distal. A total of 34 shared masses between MVs, Ds, and Ps venoms, with more shared masses between MVs and Ps venoms. The diagram on the bottom right represents the distribution of detected masses in the venom gland sections. Proximal venoms display 122 unique masses, whereas the distal section presents 71 unique masses, while 46 masses were common to both.

**Figure 4 toxins-17-00119-f004:**
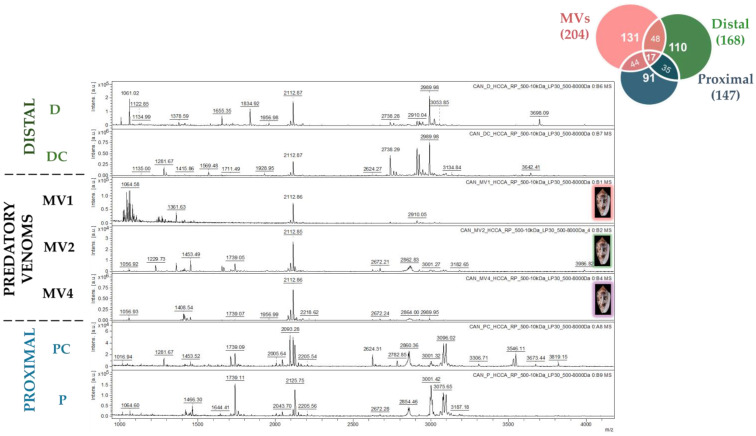
MALDI-TOF-MS spectra of the predatory milked venoms from three specimens of *Cylinder canonicus* (MV1, MV2, and MV4) aligned with dissected venoms from the distal (D, DC) and proximal (PC, P) duct sections. Matrix-assisted desorption of venom samples (dried spots) was performed at a frequency of 5000 Hz, by adding 3 × 1500 shots of a laser at a laser power (LP) of 30%. Peaks were recorded in positive reflectron mode, for the mass range 500–8000 Da in HCCA matrix. The seven mass spectra were aligned and the window was zoomed to display all detected peaks, mainly 1000–4000 Da.

**Figure 5 toxins-17-00119-f005:**
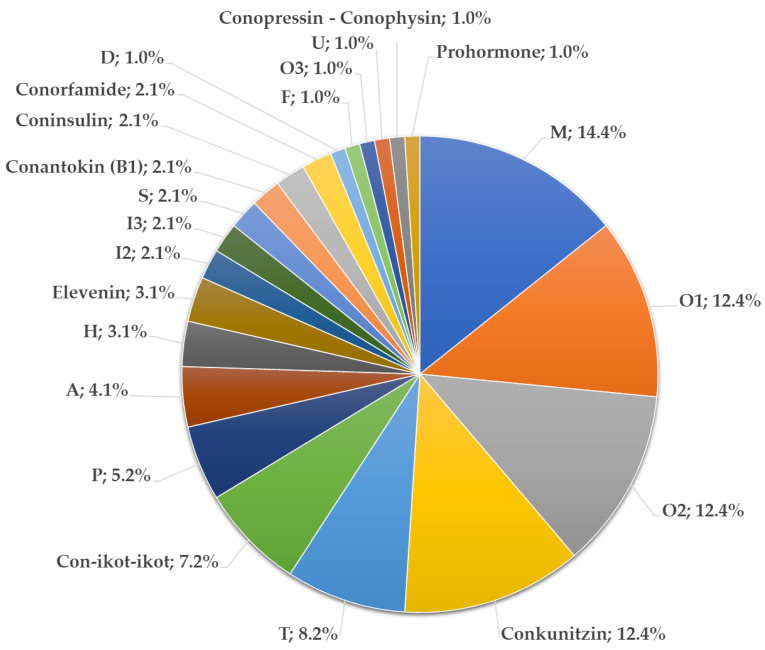
Distribution of the identified conopeptide gene superfamilies in the transcriptome of *Cylinder canonicus* venom gland. The pie chart represents proportion in number and percentage of each gene superfamilies in the transcriptome. A total of 98 conopeptides were distributed in 22 gene superfamilies.

**Figure 6 toxins-17-00119-f006:**
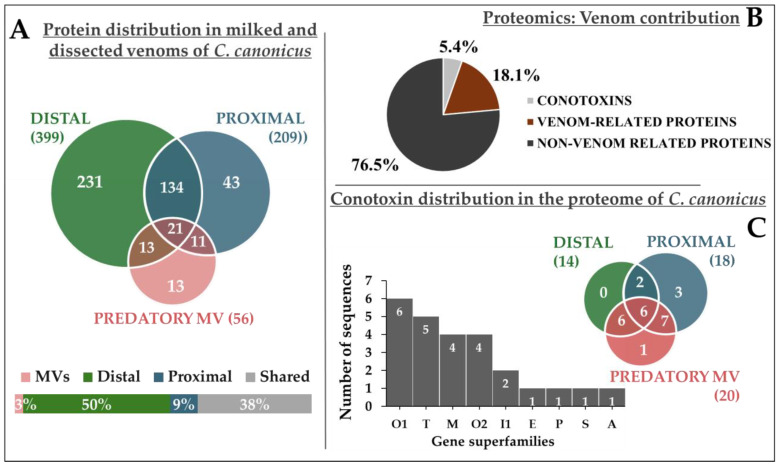
Diversity of *C. canonicus* venoms evaluated through proteomics. (**A**) The Venn diagram shows shared proteins between distal, proximal, and predatory venoms of *C. canonicus*. The figure on the bottom-left lays out size proportion of each sample and the part of shared proteins in at least two samples. (**B**) The pie chart represents total venom contribution of all venom samples assessed through blast annotation. A total of 76% of 466 proteins correspond to non-venom-related proteins (354 proteins), 18% to venom-related proteins (84 proteins), and 5% to conotoxins (25 proteins). (**C**) The histogram on the left represents the gene superfamily distribution of conotoxins validated through proteomics. Nine gene superfamilies are ordered from most to least abundant according to the number of conotoxins detected. B. The Venn diagram on the top right of the histogram represents distribution of detected conotoxins across distal and proximal venom gland sections and predatory-milked venoms of *C. canonicus*. All conotoxins, and proteins, identified through proteo-transcriptomics are presented in the Appendix A (Appendix A, Appendix A).

**Figure 7 toxins-17-00119-f007:**
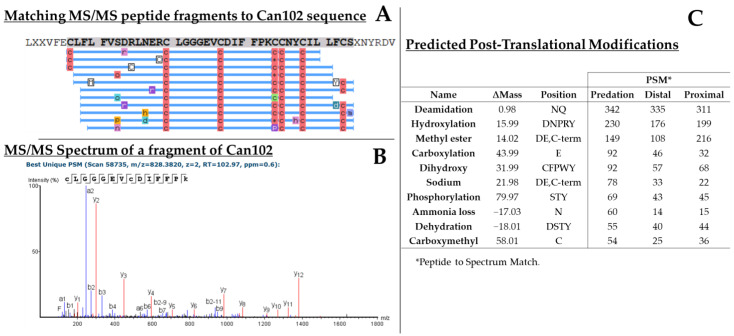
Example of proteomics results. (**A**) Protein contig containing the precursor of the peptide Can102 is detected in the predatory venom of *C. canonicus*. A total of 79 spectra were matched (not all shown in the figure for clarity), corresponding to a coverage of the mature sequence of >90%. Blue lines below the sequence represent peptide matches aligned to the protein. PTMs are also displayed on the figure with colored letters, such as c (carbamidomethylation), r (ion adducts), etc. (**B**) Example of the MS/MS spectrum automatically annotated with the b- and y-ions. (**C**) Table presenting some of the post-translational modifications (PTMs) detected in predatory-milked and dissected venoms. PTMs are quantified based on the number of peptide-spectra matches (PSM) which represent computed matches between MS/MS spectrum sequence prediction and database sequence. Of note, as they were automatically generated, some of these PTMs are likely artifacts. Only the 10 most represented PTMs are shown.

**Table 1 toxins-17-00119-t001:** Conotoxin precursor sequences identified in the venom of *Cylinder canonicus* through proteo-transcriptomics. Proteomics analyses have allowed validation of conotoxin from the experimental transcriptome database of *C. canonicus*. Here is also represented the distribution of each conotoxin in the different venom samples: the pooled predation-evoked milking and the dissected distal and proximal venoms of *C. canonicus*. Colored cells indicate that the sequence has been identified in the corresponding venom sample. For example, the O_2_-conotoxin, Can45, was found in all samples. Complete table of identified conotoxins is presented in the Appendix A (Appendix A, Appendix A).

			Proteomics Validation		
Conotoxins	Conopeptide Precursor Sequences	Gene Superfamily	Predatory	Distal	Proximal
**>Can01**	MGMRMMFIVFLLVVLATTVVSSTSGHRAFHDRNAAAKASGLVGLTDRRPQ**CC**SDPR**C**NSSHPEL**C**GGRRX	**A**			
**>Can21**	MLKMRVVLFTFLVLFPLATLQLDADQPVERYAENKQDLNPDERREIILHALGTR**CC**SWDV**C**DHPS**C**T**CC**SGX	**M**			
**>Can22**	MMLKMGVVLFIFLVLFPLATLQLDADQPVERYAENKQLLNPDERRGILLPALRKF**CC**DSNW**C**NISD**C**E**CC**YGX	**M**			
**>Can28**	MMSKLGVLLTICLLLFSLNAVPLDGDQHADQPAERLQDDVATENHPLFDPDKR**CC**DDWE**C**DYS**C**WP**CC**YGX	**M**			
**>Can33**	MKLTCMMIVAVLFLTAWTFATADDPRNGLENLFSKAHHEMKNPEASKLNKR**C**KQSGEF**C**NLIDQD**CC**DGY**C**IVFL**C**TX	**O1**			
**>Can34**	MKLTCMMIVAVLFLTTWTFATAITSNGLENLFSKAHHEMKNPEASKLNKR**C**VPYEGP**C**NWLTQN**CC**DAT**C**VVFW**C**LX	**O1**			
**>Can35**	MKLTCMMIVAVLFLTAWTLVMADDSNNGLANLFSKLRDEMEDPEGSKLEERD**C**QEKWEY**C**PVPFLSSGD**CC**IGLI**C**GPFV**C**VGWX	**O1**			
**>Can45**	MEKLTILLLVAAVLTSTQALIQGREERQKAKVRFLSKRKSTWERWWDGDCRTWNAP**C**DPSVE**CC**FGV**C**RHHR**C**VWWX	**O2**			
**>Can50**	MQKLIILLLVAAVLMSTHAMLQEKRPKEKIKFLSKRKTDAEKQQKRL**C**PDYTDP**C**SHAHE**CC**SWN**C**HNGH**C**TGX	**O2**			
**>Can53**	MEKLTILLLVAAVLMSTQALAERAGGNRLKESIKSLLKGKRSEDSRLFRD**C**TVWLAS**C**NAPSQ**CC**SAI**C**STY**C**RLWX	**O2**			
**>Can55**	MHLSLAGSAVLMLFLLFALGTFVGVQPEQITRDVDNGQLTDNRRNLQSKWKPVSLFISRRG**C**NNS**C**NEHSD**C**ESH**C**I**C**TFRG**C**GAVNGX	**P**			
**>Can61**	MSKMGAMFVLLLFTLASSQREGDIQARKTHLKRDFYRTLPRFARGCTISCEYQDNR**C**RGE**C**H**C**PGKTN**C**Y**C**TSGHHNKG**C**S**C**A**C**X	**S**			
**>Can62**	MRCLPVFVILLLLIASTPSVDARAKTRDDMSLASFHDDAKRILQILQERNACCIAKTCCRX	**T**			
**>Can99**	GPNLPHTRFHKHKRARTETKNGQLK**C**YLT**C**N**C**GPGNR**C**LGDDDIDWDHRNVKIYT**C**PSRX	**E**			
**>Can100**	QFFCPDSENDPLNCVETKGTEPACMKSKDGSYSYA**C**GY**C**GKKKES**C**FGNKVPVADYA**C**QIRKIPNP**C**GGAALX	**I1**			
**>Can101**	HSYVCGYCGKKKES**C**FGDKMPVDAYD**C**KVRNIANP**C**GGTALX	**I1**			
**>Can102**	VFECLFLFVSDRLNER**C**LGGGEV**C**DIFFPK**CC**NY**C**ILLF**C**SX	**O1**			
**>Can103**	MKLTCMMIVAVLFLTAWTFATADDPRNGLENLFSKAHHEMKNPEASKLNKR**C**-	**O1**			
**>Can104**	MKLTCMMIVAVLFLTAWTFATADDPRNGLENLFLKA	**O1**			
**>Can105**	FLTAWTFVTAVPHSSDALENLYLKARHEMENPEASKLNTRDDD**C**EPPGNF**C**GMIKIGPP**CC**SGW**C**FFA**C**AX	**O1**			
**>Can106**	NQEKHQRAKMNLLSKRKPLAERWWRWGGCMAWFGLCTKNSE**CC**SNSCDITRCELLPFPPDWX	**O2**			
**>Can107**	SDTAQLKAKDNMPLASFHGNAKQTLQMRLRNNG**CC**PGLE**CC**RFGX	**T**			
**>Can108**	MRCLPVFVILLLLTASALSVDARPKTKDDVFLSSFDDNAKSILRRIWNKRS**CC**EITFY**CC**GX	**T**			

**Table 2 toxins-17-00119-t002:** Distribution of cysteine frameworks detected within the proteo-transcriptome of *Cylinder canonicus* with their potential pharmacological family.

Cysteine Framework	Cysteine Pattern	Gene Superfamilies	Pharmacological Families
**0**	0 C	B (1); H (1); O1 (1); Conorfamides (1); Con-insulins (1); Pro-hormones (1)	
**I**	CC-C-C	A (4)	α, ρ
**III**	CC-C-C-CC	M (10)	α, ι, κ, µ
**V**	CC-CC	T (8)	ε, µ, τ
**VI/VII**	C-C-CC-C-C	O1 (14); O2 (11); O3 (1); M (2); U (1)	δ, γ, κ, µ, ω
**VIII**	C-C-C-C-C-C-C-C-C-C	S (2)	α, σ
**IX**	C-C-C-C-C-C	P (5)	ND
**XI**	C-C-CC-CC-C-C	I2 (2); I3 (2)	ι, κ
**XIV**	C-C-C-C	E (1); M (1)	α, κ
**XV**	C-C-CC-C-C-C-C	O2 (1)	ND
**XX**	C-C-C-C-CC-C-C-C-C	D (1)	α,
**XXII**	C-C-C-C-C-C-C-C	M (3)	ND
**XXIV**	C-CC-C	Con-insulins (1)	
**Unclassified**	/	F (1); I1 (3); O1 (1); O2 (3); Con-ikot-ikots (8); Conkunitzins (12); Conopressins-Conophysins (1); Elevenins (3)	

α: nicotinic acetylcholine inhibitors; κ: voltage-gated potassium channel blockers; µ: voltage-gated Na channel antagonists/blockers; ρ: α1-adrenoceptors; δ: voltage-gated Na channel agonist (delayed inactivation); γ: neuronal pacemaker (cation currents); ω: voltage-gated Ca channel blockers; σ: serotonin-gated ion channels 5-HT3, ι: voltage-gated Na channel agonists (no delayed inactivation); ND: Not Determined [5].

## Data Availability

Our raw sequencing reads were deposited to the NCBI database. The accession number for our transcriptome data is PRJNA1222818.

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
