# Peer review of "Proteo-Transcriptomic Analysis of the Venom Gland of the Cone Snail Cylinder canonicus Reveals the Origin of the Predatory-Evoked Venom"

_toxins, 2025, doi:10.3390/toxins17030119_

Round 1

Reviewer 1 Report

Comments and Suggestions for Authors

Overall this is nicely done study examining the venom composition of Cylinder canonicus.  The results are not too exciting, essentially describing new variants of conotoxins and ecological concepts that have been described earlier on multiple occasions. However, the study is carefully carried out, the results have been carefully analysed and the paper is well written and logically structured. I am supporting acceptance of this manuscript, provided the authors consider the following points:

Line 291: explain how ‘dominant’ superfamilies were identified. Where the contigs/reads quantified systematically? Are the numbers shown in Fig 5 based on number of unique contigs or on absolute quantifications of reads/contigs ?

Section 3.4: For the bottom up proteomics part, please comment on coverage.  What were the criteria used to assign the observed (unique?) peptides to proteins or families?

In the table on page 11- it doesn’t seem to have a descriptor, heading or footnote – what is the meaning of the numbers? Does “1” mean one instance was found? If so what is the difference between the 3 instances found for Can62?

Figure 7 is not mentioned in the text.

Line 411-413: I am confused by the notion that carbamidomethylation is a PTM. Does this refer to Cysteine alkylation during sample preparation? If so it does not qualify as a PTM and should not be mentioned here. Likewise, deamidation (of Asn ? Gln ?) is usually a spontaneous reaction, that could very well be due to sample handling-I would not consider this a PTM (same as oxidation (of what? Cysteine? Methionine?). Some others could be mass spec artefacts (loss of ammonia). Bromination has been observed in other conotoxins but is not mentioned at all (was it looked at) ?

This section needs more work.  Please also detail which residues were affected by the observed modifications and which conotoxins were affected?

The RNAseq data should be made publicly available, please submit the reads to NCBI short read archive or ENA. Provide accession codes.

Figures 2+3 need improvement and a little care. The Chromatograms lack clear axes labels and units (time ?). While this may seem obvious to the initiated it is poor scientific practice. Some axis labels are missing in Fig 2. On the y-axis what is meant by % ? explain. What is the meaning of the different colors ?

 Minor:

Line 38: delete ‘as’

Line 48: rephrase to “cone snail venoms” or “conus venoms”

Line 82 understand instead of understanding

Line 400: spectra instead of spectrums

Reviewer 2 Report

Comments and Suggestions for Authors

This manuscript characterizes the origin of the predation-evoked venom in one type of cone snail. The overall experimental design and analysis are solid. The reviewer has no further comments.

Author Response

We thank the reviewer for his appreciation of our work.

Reviewer 3 Report

Comments and Suggestions for Authors

Reference:  “ Uncovering the Origin of the Predation-evoked Venom in the  Mollusk-hunting Cone Snail, Cylinder canonicus”. Toxins, 2025.

General comments:  In the study submitted for publication in TOXINS, the authors characterized the venom of the snail Cylinder canonicus, a species that hunts other molluscs, worms or even fish, collected on the Island of Mayotte, located in the Indian Ocean, Southeast Africa. Integrated proteomics and transcriptomics studies were carried out, which identified 108 conotoxin sequences in 24 gene families, with a predominance of the O1, O2, M and conkunitzin families (potassium-channel pore-blocking toxins). Analysis of the venom- producing apparatus suggests that the venom originates from the distal duct. The authors discuss the importance of the data described in understanding venom-species biology and possible biotechnological applications in obtaining new drugs. After reading the text, I forward pertinent suggestions and questions, which if answered could make a revised version more complete and attractive.  

Specific Comments:

1- Setence written among lines 26 to 28  … These carnivorous predators feed on fish, mollusks or worms. They produce and use a complex venom mixture intended to paralyze small prey or to deter potential predators for defensive purposes.  Please include one or more references.

 2-  Please correct sentence written between lines 31 and 32  …  structured peptides called conopeptides (one or less disulfide bond). The result of subtraction is zero . Than re-write the sentence to …. with one or without disulfide bond !

3- Line 45 …found in the proximal section. Proximal section of what? Please complete the phrase   … proximal section of venom producing gland ?? or venom apparatus tissue???

4- Line 46 ….compared to the distal section. The same commentary as above.

5- Sentence written among lines  53 to 64, in my opinion could be part of material and methods, since is descriptive of techniques used in the research, and should be removed of introduction.

6- In the lines 80 and 81 …by combining mass spectrometry, transcriptomics and proteomics.  Better change to    …. by combining transcriptomics and proteomics analyses. Details of methodology will be shown in M/M.

7- In lines 86 to 92  … Specimens of Cylinder canonicus were collected in February 2023. Cone snails were then acclimated and maintained in marine aquariums in the laboratory. Any special reason to wait more than one year to analyze the venom and venom gland?   Wouldn't acclimatization in the laboratory be able to reduce the venom's virulence? why not collect the venom just a few days after collecting the snails, maintaining the wild characteristics of the sample?

 8- In lines 90 and 91 …. Milked venom was obtained from three different C. canonicus specimens. Is this number o specimens enough to a representative sample for proteomic analyses? What is described in literature?    

9- In lines 91 and 92 … using previously described methods [16]. Some detail of venom obtaining could be pointed!  

10- Text pointed among lines 91 to 96 …. and venom gland extracts (sections) and tissues (RNA) were retrieved from two other specimens. A whole venom gland was used for mRNA extraction, and a second venom gland was dissected into four sections (distal, distal-central, proximal-central and proximal) before extracting the venom content for mass spectrometry and proteomic analyses. Is the number of sample (one) used to obtain venom-gland and mRNAs enough to analyses? Shouldn't the authors have used a more significant number of samples to guarantee a more reliable analysis and representativity? 

11- In lines 97 to 100 authors described the obtaining of what they say predatory venom. In reference 16, the methodology used to collect what they say milked venom is described. But the differences are subtle. How can we be scientifically certain that the methodologies really represent biological and molecular differences in the samples? As a reviewer, I don't feel safe. Of course de extract venom-gland should be more contaminated with tissue components.  

12- In the lines 102 and 103 … One Cylinder canonicus specimen was dissected on ice to extract the venom gland in form of a long and thin duct. It would be very interesting, almost mandatory, for the authors to show a photographed image of what they believe to be the venom-producing gland. It is the central subject of the study and incorrect collection can mask the entire interpretation of the study. 

13- Line 103 … TRIzol regent...please change to … reagent 

14- Between lines 117 to 132 authors indicated electronic platforms used to their studies as VenUM in line 119 , BLAST+ v2.16.0+  line 127 , Swiss-Prot 127 r2024_03 databases lines 127 and 128, ConoDictor v2.4.1 line 129, and Jalview v.2.11.4.0 and ClustalW v2.1 line 131. For all these platforms and programs, authors should indicate the updated electronic address in accordance with the scientific rigor required by TOXINS. 

15- In lines 171 to 173 authors report … Milked predatory venom of three specimens of Cylinder canonicus, as well as dissected venoms of distal (D and DC) and proximal (PC and P) sections of the venom gland were prepared prior to proteomic analysis as described. My question is differences in contend of molecules between two “venoms” could not be caused by methodology used. As dissected venom could be tissue extracted components not present in “milked venom”?  

16- Regarding the data shown in figure 2 that indicate variability between the venoms collected from 3 individuals of the same species of C. canonicus. Although the technique used is very sensitive, an SDS-PAGE gel using a linear gradient of 3-20% polyacrylamide, under reducing conditions and stained with silver would be more enlightening in my opinion! And could shows since low to high molecular mass components. 

17- Those who work with the biochemistry and cellular biology of venoms always collect samples from several individuals, precisely to use a mixture of venoms to eliminate individual variations in the samples. This seems to me to be a routine among living beings. 

18- Still on figure 2, any special reason for the authors to call MV4 one of the samples instead of MV3? 

19- Finally, I would like the authors to indicate some interpretation for the variations identified, and whether this represents any biological importance beyond physiological variations such as different ages, different sexes if applicable, immediate feeding, among others. 

20- Regarding the data shown in figure 3, which identify different compositions between the venoms collected from different parts of the venom-producing apparatus of the individuals studied. Initially, it would be interesting if the authors showed a photo micrograph of this venom-producing device instead of a schematic shown by drawing. 

21- Couldn't the data shown be just a function of a dilution factor and difficulty in obtaining samples? 

22- Any special reason the authors didn't collect venom from the venom bulb? 

23- Do the authors have any idea of ​​the region where the venom is initially produced? Could it be the Venom Bulb?  If the answer is yes, then this region should also be analyzed. 

24- Without a doubt, a histological analysis of the studied regions could show significant differences, such as secretory epithelium and muscles for eliminating the venom! What could explain such as differences described.

 25- If we carry out a comparative analysis between the venoms collected in Probocis (let's say final crude venom), with the other venoms collected from other fractions of the venom-producing apparatus, we will identify that in the Probocis venom there are components in higher concentration with molecular masses smaller than the intermediate region of the other fractions, but also components with larger molecular masses. Any logical explanation from the authors for these differences?

 26- For samples with lower molecular masses, the logical explanation would be proteolytic processing. But what about the case where the concentrations of components with larger molecular masses increased?

 27- In both figure 3 and figure 4, I missed checking whether there are molecules with larger masses. This could explain the existence of one or more precursors that could be proteolytically cleaved throughout the secretory process. Hence, SDS-PAGE in a 3-20% linear gradient could improve the data. Do the methods used allow the identification of molecules with larger masses? 

28- About figure 5, where the authors portray their results in transcriptomics. It would be interesting if they added what the different colors represent in the figure legend! 

29- Still on figure 5, it would be interesting if the authors showed biochemical and functional data of the identified families instead of A, B, D, F, H, I2, I3, S, U families.  As indicated in cases for coninsulins, elevenins, conopressins-conophysins, conorfamides and prohormones. 

30- In figure 6, the authors need to standardize the colors shown in the diagram that does not show the light blue colors of the proximal and gray regions of the shared data. 

31- The title of section 3.4 … 3.4. Proteomics: characterization of the predatory-evoked venoms and dissected venom duct sections. It should come before section 3.3 which covers transcriptomics. The authors better switch section 3.4 to venomics, since they mix the proteomics and transcriptomics analyzes shown previously! 

32- Still on the data shown in figure 6, the authors describe that the vast majority of the molecules described, (76%) correspond to non-venom related molecules and only the rest (23%) would be venom-related.   … 76% of 466 proteins correspond to non-venom related proteins (354 proteins), 18% to venom-related proteins (84 proteins) and 5% of conotoxins (25 proteins). Is this not worrying and does it suggest that the venom/sample collection method should be improved, since the vast majority of components are venom-tissue apparatus contaminants with proteases that could degrade venom components and mask future interpletations? 

33- Figure 6 could be changed by placing the data shown in B at the top, and those shown in C at the bottom of the figure, since the data shown in B are preliminary to those shown in C. 

34- Furthermore, the authors could combine the data in figure 6 B into a single legend and not separated as they are. 

35- In my humble opinion, Table 2 does show data on the identifications in venoms of molecules with enormous biological potential, such as toxins that interact with cellular ion channels and all the implications that this may bring. Therefore, as this part is the most important and attractive for readers from both a pharmacological and biological point of view, I missed further exploration of the identified data.

 36- The authors could have shown some data on three-dimensional structures by docking, seeking to explore the inhibitory potential with other toxins already described and which act on the same ion channels. This would undoubtedly increase the attractiveness of the results! Thinking about biotechnological applications as postulated along of text. 

37- Dear authors, in my humble opinion the data shown in figure 7 are speculative and do not add great knowledge to the area, without additional experiments. You could use a figure from the manuscript for more detailed analysis of the Knottins identified and shown in table 2. Examples such as three-dimensional structures based on other toxins already described, docking seeking inhibitory or agonistic potential of target ion channels, among other examples. 

38- For example, biotechnological applications of the identified molecules could be justified by patch-clamp analyzes and electrophysiology assays, using the obtained fractions or even crude venom for a start. 

39- Likewise, tests of the biological activities of the fractions obtained or crude venom on cells in culture could shed some light on future biotechnological and pharmacological applications. 

40- Finally, I suggest that the authors change the manuscript title to something like: Transcriptomic and proteomic analyzes of the venom-producing gland and crude venom of the snail Cylinder canonicus.  

Round 2

Reviewer 3 Report

Comments and Suggestions for Authors

Dear colleagues from the MDPI/TOXINS Editorial Board. In this response letter, the authors were extremely elegant and competent in their responses.  The answers were clear, clarifying and non-aggressive. The answers made me more informed about the subject discussed throughout the manuscript. They also made the manuscript more complete from a scientific point of view, and with the modifications made throughout the text, the authors incorporated many data that were missing in the first version. Therefore, it is my opinion that this new corrected version can be accepted for publication in TOXINS, as it brings a lot of pertinent information in the area, which will certainly attract the attention of other researchers.  Congratulations to the authors for the good work. 

Author Response

Thank you.